# Learning Engagement and Professional Identity Among Pre-Service Teachers: The Sequential Mediating Role of Adaptability and Self-Concept

**DOI:** 10.3390/bs15070881

**Published:** 2025-06-27

**Authors:** Xiaoli Ye, Tingting Cheng, Wei Yang

**Affiliations:** Institute of Higher Education, Anhui University, Hefei 230093, China

**Keywords:** pre-service teachers, learning engagement, adaptability, self-concept, professional identity

## Abstract

The fostering of professional identity among pre-service teachers has emerged as a pivotal research focus in contemporary pedagogical studies. Significantly, learning engagement is a core component of pre-service teachers’ training during their education stage. A survey involving 632 pre-service teachers from China was conducted. It employed a sequential mediating model to explore how learning engagement relates to professional identity within the potential roles of adaptability and self-concept. The results indicated that pre-service teachers’ learning engagement was significantly related to professional identity. Specifically, adaptability and self-concept were not only independent mediators in the relationship between learning engagement and professional identity but also sequential mediators. The empirical evidence demonstrates that pre-service teachers who have a high degree of learning engagement are inclined to exhibit greater adaptability and a stronger self-concept, which can strengthen their professional identity.

## 1. Introduction

Teacher professional identity is generally regarded as a teacher’s understanding and perception of their own role ([25]). This concept encompasses multiple dimensions such as a teacher’s cognition, beliefs, emotions, motivations, and attitudes towards their professional role, and it is always in a state of dynamic development and change ([10]). Teacher professional identity has consistently been a prominent focus in teacher education research, and its significance has been demonstrated from various perspectives. For instance, [34] ([34]) emphasized the importance of constructing professional identity in teacher education practice, stating that the mission of teacher education is to help pre-service teachers reshape their identities. [79] ([79]) found, through a review of relevant research over the past decade, that there is a significant correlation between pre-service teachers’ classroom management ability and the strength of their professional identity. Additionally, the impact of professional identity on teachers’ career development has also been discussed. Research indicates that professional identity plays a crucial role in shaping educators’ sense of belonging and commitment to the teaching profession ([59]). It not only influences teachers’ job satisfaction and well-being but also relates to their career development and attitudes towards educational reforms ([53]). Specifically, educators with deep-seated professional identity have a tendency to establish positive behavioral and attitudinal connections with their profession. They can effectively navigate adverse educational working conditions and significantly mitigate feelings of job burnout and turnover intentions ([43]). On the contrary, a decline in the sense of a teacher’s professional identity may adversely affect various dimensions such as work commitment, professional development, mental health, and teaching quality ([91]). Evidently, whether professional identity is stable or not has become an important indicator for evaluating the stability of teaching staff and functions as a crucial factor in ensuring educational quality. The OECD report indicates that teachers’ professional identity plays a fundamental role in enhancing educational quality and fostering long-term career development ([87]). Another empirical study on Chinese teachers further demonstrates that professional identity indirectly ensures the stability of educational quality by strengthening teachers’ professional commitment ([100]).

The significance of professional identity is particularly prominent in the context of the global teacher shortage crisis. At present, many countries are facing severe crises in teacher recruitment and retention. For instance, a recent government report in Australia predicts a shortage of over 4000 secondary school educators by 2025 ([22]). In England, the recruitment of secondary school teachers has failed to meet its targets for several consecutive years ([35]). To attract more teachers to the profession, governments around the world have formulated targeted teacher education programs (such as the “Teach for Australia” program in Australia and the “Teach First” program in the UK) and begun to focus on the training of pre-service teachers. Against this backdrop, cultivating pre-service teachers with a strong sense of professional identity has emerged as a key breakthrough in addressing the teacher shortage crisis. Research indicates that the professional identity formed by pre-service teachers during their teacher education stage can directly predict their future career choices and retention intentions ([43]; [40]). A weakened sense of professional identity may lead to a higher risk of attrition by reducing self-efficacy ([27]). Pre-service teachers, as a crucial component of the future teaching workforce, demand targeted support for shaping their professional identity, which is vital for the sustainable development of the teaching profession ([13]).

In China, normal universities play a pivotal role in pre-service teacher education. Moreover, some comprehensive universities with a tradition of teacher education also offer degree programs for pre-service teacher education. However, pre-service teacher education in Chinese universities generally faces similar issues to those in other countries around the world, including insufficient attractiveness of pre-service teacher training programs ([102]) and a decline in the retention rate of pre-service teachers ([23]). These issues serve as a reminder that the pre-service teachers’ professional identity should receive widespread attention. Professional identity not only directly affects whether pre-service teachers will choose to enter the teaching profession after graduation ([45]) but also relates to how long they will remain in the profession after becoming teachers ([59]). It is worth noting that during the pre-service teacher training process, there may be significant differences in professional identity between the education stage and the initial practice stage. This study focuses on the professional identity of pre-service teachers during the education stage, which is mainly based on the typical feature of the Chinese teacher education system of “emphasizing theory over practice”. Specifically, in the Chinese educational context, pre-service teachers usually need to spend at least three years of a four-year program on subject-specific and teaching skills courses, while the practice stage typically lasts only 1 to 3 months ([103]). During the education stage, learning engagement, as a crucial indicator for assessing professional development among pre-service teachers, is directly linked to their accumulation of knowledge, enhancement of skills, and formation of professional qualities ([16]). Consequently, this study identifies learning engagement as an important predictive variable influencing professional identity. At present, various investigations have revealed the correlations tying pre-service teachers’ learning engagement to professional identity. For example, research conducted by [53] ([53]) found that pre-service teachers’ learning engagement is closely associated with their professional identity. When investigating elements that shape professional identity, it is essential to consider both learning engagement and psychological factors. Among these, adaptability, as an important psychological characteristic of individuals when facing new environments and tasks, is highly likely to exert a far-reaching influence on professional identity ([78]; [69]). Furthermore, self-concept has been considered another critical psychological factor in fostering teacher professional identity, which can impact professional identity considerably ([104]; [10]). Several studies have conducted certain explorations on the relationships among pre-service teachers’ learning engagement, adaptability, self-concept, and professional identity. Previous studies have mostly investigated the influences of these independent factors on professional identity separately, and few studies have endeavored to systematically examine the relationships among these factors and how they interact to affect pre-service teachers’ professional identity.

Building on these considerations, this study aims to systematically explore the correlation mechanism among pre-service teachers’ learning engagement, adaptability, self-concept, and professional identity among Chinese pre-service teachers. We introduce adaptability and self-concept as two crucial variables to explore the mechanisms through which learning engagement shapes professional identity during the pre-service teachers’ training phase. Specifically, we have established a hypothesized model suggesting that adaptability and self-concept function as mediators through which learning engagement fosters professional identity among pre-service teachers. The current study performed a statistical analysis on the relevant data collected via questionnaire surveys, aiming to investigate the relationship between learning engagement and professional identity among Chinese pre-service teachers. By addressing these issues, this inquiry contributes to an enhanced theoretical cognition of the correlation between pre-service teachers’ learning engagement and professional identity while providing meaningful and actionable practical guidance for the cultivation of Chinese pre-service teachers.

## 2. Literature Review and Research Hypothesis

### 2.1. Learning Engagement and Professional Identity

Learning engagement is characterized as the positive psychological condition exhibited by learners across the entire learning journey ([83]). It is characterized not merely as a temporary and particular phenomenon, but instead as a continuous and pervasive cognitive state of general significance. Evidence suggests that learning engagement serves not only as a crucial factor in motivating learners to develop academic motivation and achieve their educational goals ([31]) but also acts as a key indicator capable of reliably predicting academic achievements ([75]). Typically, learning engagement is conceptualized as comprising various components. In [30]’s ([30]) model, learning engagement is categorized into two primary components: behavioral engagement, which encompasses aspects such as class participation and school attendance, and emotional engagement, which includes elements like school identification, a sense of belonging, and the significance attributed to academic pursuits. [31] ([31]) expand this framework by introducing cognitive engagement, positing that learning engagement should incorporate cognitive-level involvement in understanding learning tasks, strategy utilization, and self-regulation. This classification enhances our understanding of learning engagement and provides specific internal indicators for each subtype, thus facilitating a more comprehensive assessment of the learner’s state throughout the educational process.

In teaching and teacher education, scholars have defined the concept of “teacher professional identity” from different perspectives. [10] ([10]) pointed out that teacher professional identity is usually understood as teachers’ cognition of their role as educators and their social recognition as teachers, emphasizing that this is a role positioning formed through the interaction between individuals and society. The OECD report defines teacher professional identity as a collection of beliefs, emotions, and perspectives about themselves and their role as teachers ([87]), focusing on the internal beliefs and emotional experiences of professional identity. Combining various explanations of teachers’ professional identity, it can be seen that it is a dynamic construction process gradually formed through continuous reflection and adjustment under the joint influence of social interaction and individual psychological factors ([53]). In fact, as a multi-dimensional concept of identity, the formation of teachers’ professional identity is influenced by many factors ([9]). These factors include personal beliefs and emotions ([25]), motivation ([103]; [15]), and other internal psychological traits. They also encompass external situational factors such as social background and interaction with others ([34]), school institutions ([80]), and working conditions ([70]). Similarly, the measurement tools for teacher professional identity also show different emphases. Currently, existing research has investigated the relevant indicators of the composition of teacher professional identity, including professional commitment ([53]), job satisfaction ([14]), self-efficacy ([2]), and motivation level ([43]), etc. However, cross-cultural studies have shown that the structure of professional identity may vary due to differences in educational systems and cultural values ([41]). For instance, the dimensions proposed by Chinese scholars, such as professional value and role identity ([96]), better reflect the influence of collective orientation and social expectations on career choices in the Chinese cultural context, while the Western framework ([53]) focuses more on individual teaching commitment. Given that the subjects of this study are pre-service teachers in China, this study adopts the framework proposed by ([95]), which includes four dimensions to measure the professional identity of pre-service teachers: professional aspiration, professional will, professional value, and professional efficacy. On the one hand, this model incorporates the “professional value” dimension, which is localized and aligns with the characteristics of Chinese teacher education, emphasizing social responsibility. On the other hand, this model has been verified through empirical research and has good reliability and validity for the pre-service teacher group in the Chinese cultural context, making it a standardized and effective tool for assessing the professional identity of pre-service teachers ([95]).

The construction of a teacher’s professional identity is based on personal learning experiences, which include feedback from teaching practice, interactions with educational peers, and personal self-reflection ([17]). This process can be explained by the Social Cognitive Career Theory (SCCT). This theory originated from Albert Bandura’s self-efficacy theory and general social cognitive theory ([57]). Since its proposal by [58] ([58]), SCCT has developed into a systematic career theory, emphasizing that an individual’s career path is determined by the dynamic interaction of multiple factors. Among them, situational variables influence the formation of career interests and career choices by shaping an individual’s learning experiences. Specifically, the learning experiences accumulated by an individual in the environment interact with environmental variables. When learning experiences change, an individual’s career cognition also adjusts, thereby promoting dynamic changes in the career development path. Existing research has also confirmed the crucial role of learning engagement in the formation of pre-service teachers’ professional identities. For instance, a recent study in Ghana ([1]) also confirmed this finding, indicating that previous teaching and learning experiences are a significant driving force for their choice to become teachers and can significantly enhance their professional aspiration. In Germany, [66] ([66]) conducted a study on 490 prospective teachers at the University of Hamburg, highlighting that the knowledge gained by pre-service teachers in university courses plays a crucial role in the formation of their professional will. These two studies, respectively, reveal the impact of learning engagement on the professional identity of pre-service teachers from different perspectives. For pre-service teachers in China, the current teacher education programs mostly adopt a training model that combines theoretical courses with practical components. In the curriculum system, the imparting of theoretical knowledge and the training of subject-specific professional skills take up a relatively large proportion ([103]; [59]). Insufficient learning engagement may affect the effective transformation of classroom knowledge into teaching practice abilities, thereby hindering the formation of their professional identity. Research indicates that active learning engagement is conducive to clarifying career directions ([54]), while the opposite may dampen career enthusiasm ([99]). Based on the above theoretical and empirical evidence, we propose the following hypotheses:

**H1:** 
*Pre-service teachers’ learning engagement positively predicts their professional identity.*


### 2.2. The Mediating Role of Adaptability

Adaptability, as defined by [63] ([63]), is the potential of an individual to make appropriate cognitive, behavioral, or emotional adjustments when confronted with uncertainty and novelty. This concept has been extensively explored in the domain of education, especially focusing on aspects such as academic adaptability and psychological adaptability among freshmen at university. For example, [92] ([92]) indicated that university students’ academic adaptability is a crucial element in predicting their academic achievements. [64] ([64]) investigated the psychological adaptability challenges faced by university students during attachment to parents and separation-individuation. Their findings suggest that secure parent–child attachment relationships enable university students to adapt effectively to university life. This study mainly focuses on academic adaptability and psychological adaptability. Academic adaptability is a psychological and behavioral process, referring to the ability of students to adjust their learning strategies to adapt to the environment and learning needs ([18]). Psychological adaptability refers to an individual’s ability to integrate psychological resources to cope with challenges or stress ([77]).

Previous studies suggested that individuals’ adaptability exerts a remarkable influence on their professional identity ([48]; [97]). Individuals with inferior adaptability tend to demonstrate uncertainty in career planning, feel perplexed about their future directions, and lack clear goals and motivations. Conversely, those with superior adaptability are capable of confronting various challenges in the process of learning or career planning more proactively, setting clear goals, and formulating feasible action plans. [69] ([69]) discovered that good adaptability has a positive impact on the construction of professional identity, and professional identity is not necessarily fixed but may be transitional, constantly varying, and fluctuating based on the outcome of interaction with adaptability in specific contexts. Research conducted by [25] ([25]) revealed that adaptability is regarded as a crucial factor in maintaining stable and unstable professional identity. Individuals who possess good adaptability can more effectively adjust their mindset and behavior when encountering changes in the professional environment, thereby preserving the stability of their professional identity. [37] ([37]) also found that adaptability shows a positive correlation with job-seeking self-efficacy and employment status. It further shows its role in promoting professional identity.

Learning engagement is a crucial factor for university students in adapting to campus life. [94]’s ([94]) research revealed that learning engagement can positively predict students’ psychological adaptability and academic performance. [36] ([36]) recognized three dimensions of student engagement: behavioral, emotional, and cognitive. These three dimensions are positively correlated with students’ adaptability and academic achievements. Moreover, studies have indicated that the extent of university students’ engagement in learning is positively correlated with their adaptability, and this adaptability subsequently enhances their academic performance ([63]). [24] ([24]) further substantiated the significant correlation between learning engagement and adaptability. Students having a high degree of learning engagement are inclined to exhibit stronger adaptability, enabling them to cope with academic challenges, interpersonal relationships, and environmental changes more effectively. Based on the existing research findings, the degree of learning engagement is strongly linked to university students’ academic or psychological adaptability. Learning engagement not only positively predicts the adaptability of these students but also facilitates the formation of their professional identity through enhancing adaptability. Therefore, the subsequent hypothesis is put forward:

**H2:** 
*Adaptability might mediate the relationship between pre-service teachers’ learning engagement and professional identity.*


### 2.3. The Mediating Role of Self-Concept

Self-concept is defined as a person’s subjective evaluation of their own abilities and values, which is modeled by experiences and how they perceive their environment. [86] ([86]) divided self-concept into academic aspects and non-academic aspects. Academic self-concept involves students’ self-awareness about their academic capabilities, which is the students’ anticipation and judgment of whether they can succeed and master a specific academic task. A substantial body of research has investigated academic self-concept along with its influencing factors. Research has indicated that university students’ academic self-concept significantly affects their academic achievement ([62]), academic engagement ([12]), and professional selection ([26]). On the basis of the existing related research on the internal structure and classification of self-concept, our study further categorizes it into two distinct types: academic self-concept and professional self-concept. Professional self-concept is defined as an individual’s knowledge, adoption, and reflection of the specific characteristics, standards, and skills associated with a profession ([4]).

Researchers thoroughly explored the correlation between self-concept and professional identity across various fields. Several studies have demonstrated that professional identity is closely intertwined with teachers’ self-concept or self-image ([50]; [68]). [81] ([81]) discovered that academic self-concept can enhance the pre-service teachers’ professional identity through teachers’ efficacy. Additional research revealed that academic self-concept is positively linked to students’ professional aspirations ([51]), further indicating the significant influence of self-concept on individuals’ professional selection and professional identity. [72] ([72]) discovered that pre-service teachers’ professional knowledge is strongly correlated with their academic self-concept and begins to show differentiation in the initial phases of teacher education.

Furthermore, an extensive body of research has delved into the connection between learning engagement and self-concept, although a unanimous conclusion on the specific correlation mechanism between the two has not yet been established. For example, [3] ([3]) discovered a positive correlation between student engagement and creative self-concept, emphasizing that learning engagement is positively correlated with the development of students’ creativity and the improvement of self-cognition. [84] ([84]) suggested that heightened learning engagement fosters a clearer academic self-concept, which in turn enhances academic performance. [44] ([44]) also revealed that student engagement is correlated with self-concept, and students’ extracurricular or in-class learning engagement is positively associated with their creativity, self-confidence, and enterprise. In conclusion, there is a connection between learning engagement and the formation of self-concept, and self-concept is also closely related to professional identity. Thus, the hypothesis is proposed as follows:

**H3:** 
*Self-concept might mediate the relationship between pre-service teachers’ learning engagement and professional identity.*


### 2.4. The Sequential Mediating Role of Adaptability and Self-Concept

From the above literature, it can be seen that adaptability and self-concept play a significant role in shaping the professional identity of pre-service teachers. At the same time, this study also focuses on exploring the mediating role of adaptability and self-concept in this process. What is the theoretical basis for this exploration? This study believes that the core viewpoints of social cognitive theory can provide certain support. Social cognitive theory holds that human activities are determined by the dynamic interaction among individuals, behaviors, and environmental factors ([6]). Within the framework of social cognitive theory, [8] ([8]) introduced four interrelated core concepts to explain human agency: intention, forethought, self-reactiveness, and self-reflectiveness. This theory emphasizes that humans are not merely passive organisms responding to external events but are subjects with the ability to self-reflect and self-regulate ([7]). During the self-reactiveness stage, individuals, based on personal standards, supervise and regulate their actions through the influence of self-reactiveness to adapt to the constantly changing environment ([8]). In the self-reflectiveness stage, individuals reflect on their own efficacy, the ideas of action, and the rationality of action, evaluate their abilities and behaviors, and determine future goals. Therefore, social cognitive theory can provide theoretical support for the sequential mediating hypothesis model explored in this study. The self-concept of pre-service teachers, as a reflective cognition and evaluation of themselves, plays a crucial role in their continuous adaptation to the learning process of pre-service teachers. Specifically, before officially becoming teachers, pre-service teachers continuously learn professional knowledge and participate in educational practice projects. They adapt to the demands of the training field by adjusting learning strategies and adapting to educational practices and other adaptive behaviors. Subsequently, these behavioral experiences are integrated into the self-concept system by individuals, forming a cognitive evaluation of their professional abilities. Eventually, this self-concept, strengthened through reflection, prompts individuals to establish a stable professional identity. In the long run, this behavior pattern based on self-concept regulation will have a profound impact on the development of their professional understanding and identity. In summary, whether in the educational setting or the future school environment where they will work, the specific behaviors and subjective initiative of pre-service teachers are always shaped by the interaction of personal, behavioral, and environmental factors. Therefore, during the educational stage of pre-service teachers, emphasizing the development of adaptability and self-concept is conducive to forming a more complete professional identity.

Furthermore, another theoretical support path for the sequential mediation hypothesis model constructed in this study comes from the empirical findings of previous related research. Empirical studies have also supported the significance of the dynamic interaction between adaptability and self-concept. [32] ([32]) pointed out that students with strong adaptability can enhance their academic self-concept by leveraging their character strengths, thereby promoting their professional development. [33] ([33]) noted that self-regulated learning enables college students to recognize their own abilities and limitations. This awareness of their academic capabilities strengthens their academic self-concept and serves as a crucial foundation for their professional positioning. Additionally, related research further indicates that the adaptability demonstrated by college students in academic adaptation, stress regulation, and interpersonal communication has a positive impact on their self-concept and can significantly predict their academic achievements ([101]; [47]).

Increasing evidence indicates that university students’ learning engagement significantly influences their professional identity. The mechanism can be understood from two primary perspectives. First, examining the inherent structural connection that bridges learning engagement and professional identity, which encompasses how these two constructs interact with one another. [76] ([76]) have found that learning engagement is a key predictor in the development of professional identity. University students cultivate their professional identities through vocational learning; conversely, their established professional identities also impact their levels of learning engagement. The second perspective focuses on examining the mediating role of psychological outcomes derived from university students’ learning engagement in shaping their professional identity. [20] ([20]) and [61] ([61]) have explored this phenomenon across various professional contexts, demonstrating that enhanced learning engagement can significantly bolster students’ professional identities by reinforcing psychological items such as academic self-efficacy and psychological resilience.

However, a comprehensive analysis of the content and subjects of existing research reveals that there are still several directions that need to be further explored in this field. First, the specific mechanisms and interrelationships among these psychological factors, as well as their combined impact on professional identity, have not been fully clarified by current research. Second, there is insufficient attention paid to the “Chinese pre-service teachers” group. The formation mechanism of professional identity among Chinese pre-service teachers has distinct local characteristics, but existing research has not adequately revealed the deep interaction logic between learning engagement and professional identity in the Chinese context, making it difficult to construct an explanatory model that conforms to the psychological development laws of Chinese college students. As a unique group embodying both the characteristics of college students and a distinct dedication to the education profession, pre-service teachers’ learning engagement may be more strongly associated with the development of professional identity. Based on this, this study focuses on the Chinese pre-service teacher group and, in combination with social cognitive theory, proposes a sequential mediation model: learning engagement can promote teacher professional identity through a two-stage pathway involving adaptability and self-concept. Consequently, we put forth the following hypothesis:

**H4:** 
*Adaptability and self-concept sequentially mediate the relationship between pre-service teachers’ learning engagement and professional identity.*


To elucidate the proposed model more clearly, Figure 1 displays a theoretical framework diagram that delineates the interrelationships among the variables.

## 3. Research Methods

### 3.1. Data Source

We selected pre-service teachers from freshmen to seniors from three Chinese universities who have established cooperative relationships with the research team to participate in the survey. Then, pre-service teachers are invited to participate in the research through a combination of online and offline methods. Online links to questionnaires with informed consent are published through the Official Learning Platform of the University, while offline paper questionnaires are distributed with the assistance of teachers from cooperative universities to explain the purpose of the research and the method of participation. At the same time, in order to ensure the randomness of the sample, the online system randomly selected student accounts and pushed questionnaires and the offline random distribution of a specified number of questionnaires in each class. The questionnaire was distributed online and offline. We ensured data secrecy and protected participants’ privacy after all inquiries that could potentially reveal personal identities, such as names, addresses, or other identifying details. This approach aimed to guarantee data submission anonymity and preclude any direct tracking.

Respondents had the right to complete the questionnaire at their convenience and could stop participating at any time. During the survey process, researchers reminded each student to answer carefully and recorded the time taken to complete the questionnaire. Ultimately, 745 questionnaires were successfully collected. Before conducting data analysis, we followed the established procedures and standards to clean the data. Firstly, questionnaires with too short completion times, those with input errors, and those with missing data were excluded. Secondly, non-randomly missing responses and cases with excessively high option repetition rates were removed. These situations were classified as abnormal or invalid data. After this rigorous cleaning process, 632 participants were selected for further analysis.

Table 1 exhibits the demographic data concerning the participants. Specifically, male participants took up 18.0%, and female participants made up 82.0%. The distribution between rural and urban samples is 52.1% and 47.9%, respectively. In total, 52.1% of the participants were public-funded pre-service teachers and 47.9% were non-public-funded pre-service teachers. Furthermore, the proportions for freshmen, sophomores, juniors, and seniors are as follows: 43.2%, 18.5%, 28.8%, and 9.7%, respectively; the average age of the research subjects is 20.08.

### 3.2. Measurement Tools

#### 3.2.1. Learning Engagement Scale

The Learning Engagement Scale was devised by drawing on existing measurement instruments and was subsequently refined and tailored to meet the requirements of the particular study context ([38]). In total, 24 items were encompassed, mainly focusing on six dimensions: course study, deep learning approach, student–faculty interaction, peer interaction, extracurricular activity, and university belonging. A 5-point Likert scale is used to assess the items, from 1 (never) to 5 (very often). The higher the score, the more suitable it is for the item. The results of a confirmatory factor analysis demonstrated acceptable fit indices (χ^2^/df = 4.93, NFI = 0.83, CFI = 0.86, RMSEA = 0.085) for the scale in this research. The Cronbach’s α coefficients for each dimension were 0.87, 0.87, 0.88, 0.88, 0.88, and 0.87. These findings suggest that this scale reveals relatively high levels of reliability and validity.

#### 3.2.2. Adaptability Scale

The Adaptability Scale was devised by drawing on existing measurement instruments and was subsequently refined and tailored to meet the requirements of the particular study context ([29]). There are 19 items in total, which were mainly related to two dimensions: academic adaptability and psychological adaptability. These items are intended to reflect the level of adaptability of pre-service teachers. A 5-point Likert scale is employed for evaluating the items, with higher scores indicating higher adaptability. The results of a confirmatory factor analysis demonstrated acceptable fit indices (χ^2^/df = 5.03, NFI = 0.90, CFI = 0.88, RMSEA = 0.08) for the scale in this research. It was found in this research that Cronbach’s α coefficients for both dimensions amounted to 0.87. We can conclude that this scale demonstrates relatively good reliability and validity.

#### 3.2.3. Self-Concept Scale

The Self-concept Scale compiled by [39] ([39]) was applied to measure self-cognition and evaluation among university students. It comprises 10 items and is specifically formulated to assess the self-concept degree. These items primarily relate to two dimensions: academic self-concept and professional self-concept. A 5-point Likert scale is employed for evaluating the items, with higher scores indicating higher levels of self-concept. Confirmatory factor analysis yielded the following results: χ^2^/df = 3.79, NFI = 0.96, CFI = 0.97, RMSEA = 0.07. The Cronbach’s α coefficients for the two dimensions were both 0.87. These findings indicate that the scale demonstrates relatively strong reliability and validity.

#### 3.2.4. Professional Identity Scale

The Professional Identity Scale was devised by drawing on existing measurement instruments and was subsequently refined and tailored to align with the specific context of this research ([95]). The Professional Identity Scale is grounded in the actual training context of Chinese pre-service teachers and has been specifically tailored for this demographic group, ensuring high relevance and applicability. There are 12 items in total, which were predominantly associated with four dimensions: professional aspiration, professional will, professional value, and professional efficacy. A 5-point Likert scale is used to assess the items, from 1 (completely) to 5 (completely agree), with a higher score corresponding to a higher level of agreement with the item. Confirmatory factor analysis yielded the following results: χ^2^/df = 3.08, NFI = 0.95, CFI = 0.97, RMSEA = 0.06. The Cronbach’s α coefficients for each dimension were 0.88, 0.88, 0.88, and 0.87. These findings indicate that the scale demonstrates relatively strong reliability and validity.

### 3.3. Statistical Analysis

The research used Amos 26.0 to assess the internal consistency and validity of the scale’s structure. Subsequently, statistical analyses were conducted on the data collected utilizing SPSS 27.0, which included descriptive statistics and correlation analysis. Finally, this study employed a predictive model through PROCESS v4.0 with the objective of exploring the relationships among pre-service teachers’ learning engagement, adaptability, self-concept, and professional identity as educators, particularly examining the mediating roles of adaptability and self-concept in this context.

## 4. Results

### 4.1. Common Method Bias Test

The measurements of the subjects in this investigation utilized various scales, which could introduce common method bias into the results. Before conducting formal data analysis, we utilized Harman’s single-factor method to evaluate potential common method bias. This approach allows for a more accurate evaluation of whether a single factor explains all observed variance. The findings indicated that in the unrotated solution, there were 13 initial eigenvalues exceeding 1, and the first common factor explained 28.98% of the variance, which falls below the standard critical threshold of 40%. Therefore, we can infer that the data collected for this study do not exhibit substantial common method bias.

### 4.2. Descriptive Statistics and Correlation Analysis

Table 2 presents a summary of the descriptive statistics and the correlation matrix. Regarding learning engagement, pre-service teachers scored highest on the university belonging (M = 3.89, SD = 0.77), showing a relatively positive learning experience. The scores on the deep learning approach (M = 3.70, SD = 0.60) and course study (M = 3.69, SD = 0.74) ranked second and third, respectively, suggesting that pre-service teachers allocated more effort to academic studies during their study period. The score on the peer interaction (M = 3.30, SD = 0.81) exceeded the theoretical median, signifying that pre-service teachers attached considerable importance to collaboration and communication among peers during learning. However, the scores on the extracurricular activity (M = 2.35, SD = 0.85) and student–faculty interaction (M = 2.98, SD = 0.92) both fell below the theoretical median, indicating that pre-service teachers had relatively low participation in extracurricular learning and a negative experience in interaction with teachers. As to professional identity, pre-service teachers scored above 4 on the professional aspiration (M = 4.36, SD = 0.63), professional value (M = 4.01, SD = 0.67), and professional efficacy (M = 4.19, SD = 0.62), suggesting a high inclination towards teaching and a clear perception of the value and efficacy of the teaching profession. Nevertheless, the score on the professional will (M = 3.62, SD = 0.89) was relatively low, indicating a potential risk in their professional perseverance. Concerning adaptability, pre-service teachers scored above the theoretical median on both academic adaptability (M = 3.79, SD = 0.61) and psychological adaptability (M = 3.45, SD = 0.70), suggesting that pre-service teachers possessed good adaptability when confronted with changes in the learning environment and psychology. However, compared to academic adaptability, their performance in psychological adaptability was less satisfactory, suggesting that they might be beset by academic stress and career planning and require further improvement in emotional management and stress coping. With respect to self-concept, the scores on the academic self-concept (M = 3.78, SD = 0.59) and professional self-concept (M = 3.67, SD = 0.81) were comparable and exceeded the theoretical median of 3, indicating that pre-service teachers had a favorable cognition and evaluation of their academic and professional planning.

In analyzing the correlations between variables, as detailed in Table 2, notable positive correlations were observed between learning engagement, adaptability, self-concept and professional aspiration, professional will, professional value, and professional efficacy. The deep learning approach had significant positive correlations with academic self-concept, professional self-concept, academic adaptability, psychological adaptability, professional aspiration, professional value, and professional efficacy (*p* < 0.01). In light of the correlation analysis results, PROCESS v4.0 can be further utilized to construct the hypothesized relationships involving variables in the subsequent study.

### 4.3. Mediation Model Test

The pre-service teachers’ learning engagement is set as an independent variable, and professional identity is set as a dependent variable. At the same time, gender, residence, grade, and cultivation type are taken into account as control variables, while adaptability and self-concept are taken as mediating variables. Based on these settings, a sequential mediation model was established. We employ model 6 from the PROCESS plug-in, created by Hayes, to calculate the mediating impact of adaptability and self-concept between pre-service teachers’ learning engagement and professional identity. By taking 5000 bootstrap samples and calculating a 95% confidence interval, the significance of the mediating effect was tested. When this interval excluded zero, it indicated a statistically significant mediating effect ([74]). As Table 3 and Table 4 and Figure 2 demonstrate, in the sequential mediation model, the R2 value of the model was 0.257, implying that the model accounted for 25.7% of the variance in the professional identity variable. The F-statistic is 30.894, with a significant F-change of 0.000. The statistical significance of the overall model suggests that the proposed sequential mediation framework effectively elucidates the interrelationships among the variables.

Table 3 reveals that pre-service teachers’ learning engagement exhibited a substantial and positive correlation with adaptability (β = 0.815, t = 25.259, *p* < 0.001) and self-concept (β = 0.379, t = 9.501, *p* < 0.001). Adaptability was positively associated with self-concept (β = 0.474, t = 13.633, *p* < 0.001). In the first stage of the regression analysis, pre-service teachers’ learning engagement was significantly related to professional identity (β = 0.426, t = 11.830, *p* < 0.001), thus affirming hypothesis 1. The 95% confidence interval of the total effect of pre-service teachers’ learning engagement on professional identity was [0.355, 0.497], and since 0 was not included in this range, the total effect was deemed statistically significant. In the second phase of the regression analysis, it was shown that adaptability plays a positive role in shaping professional identity (β = 0.231, t = 4.928, *p* < 0.001), and self-concept positively contributed to professional identity (β = 0.283, t = 5.968, *p* < 0.001).

The direct effect of pre-service teachers’ learning engagement on teacher professional identity was insignificant (β = 0.021, t = 0.425, *p* > 0.05), as the 95% confidence interval of [−0.078, 0.121] included 0. Consequently, the sequential mediating effect was a complete mediation.

The sequential mediation effect was primarily composed of the following three paths (as detailed in Table 4), with the results of the path coefficients illustrated in Figure 2:(1)Pre-service teachers’ learning engagement → adaptability → professional identity (95% confidence interval = [0.106, 0.272]). Since 0 is not included in this range, this finding validates Hypothesis 2, which predicted the significance of this indirect effect.(2)Pre-service teachers’ learning engagement → self-concept → professional identity (95% confidence interval = [0.061, 0.160]). Since 0 is not included in this range, this finding validates Hypothesis 3, which predicted the significance of this indirect effect.(3)Pre-service teachers’ learning engagement → adaptability → self-concept → professional identity (95% confidence interval = [0.065, 0.159]). Since 0 was not included in this range, this finding validates Hypothesis 4, which predicted the significance of this indirect effect.

## 5. Discussion

### 5.1. Pre-Service Teachers’ Learning Engagement and Professional Identity

In light of the outcomes of the present investigation, pre-service teachers’ learning engagement can positively predict pre-service teachers’ professional identity. As pre-service teachers’ learning engagement increases, their comprehension of professional identity also deepens ([11]; [98]). In this report, we evaluate pre-service teachers’ learning engagement from six dimensions. Pre-service teachers can actively promote professional identity by enhancing their engagement in these six dimensions. In other words, professional identity is the product of multiple processes, which occur in different settings of pre-service teachers’ learning and life ([60]). The empirical study finds a connection between several factors and professional identity. Among these, the several dimensions of learning engagement are often mentioned. [71] ([71]) proposed that by studying vocational courses, students’ academic self-efficacy can be significantly enhanced, which is conducive to establishing a strong psychological foundation and making students have greater expectations for future employment. Similarly, [89] ([89]) discovered that the deep learning approach facilitates the development of professional identity. Studies have also emphasized the interaction between pre-service teachers and their peers, the participation in extracurricular activities, and the feedback and support from teachers enable these individuals to attach importance to their performance, thereby strengthening their self-image ([21]; [82]; [93]). Previous research also showed that students with a stronger sense of university belonging will obtain greater academic achievement and learning engagement ([67]). Therefore, the formation of professional identity among pre-service teachers requires the support of universities to provide timely assistance and guidance ([28]). Enhancing learning engagement is of great significance to pre-service teachers to access professional knowledge, cultivate positive learning behaviors, achieve better learning outcomes, and actively participate in practical activities.

### 5.2. The Mediating Role of Adaptability

These findings reveal a significant association between pre-service teachers’ learning engagement and professional identity, with adaptability emerging as a potential mediator. Specifically, the more effort and time they dedicate to their academic endeavors, the greater their adaptability becomes, which subsequently enhances their professional identity. Pre-service teachers face numerous challenges within their university studies and personal lives. High levels of learning engagement encourage them to make continuous progress in their academic pursuits, manage daily life effectively, and acquire necessary skills. This study supports previous research ([36]), which confirmed that pre-service teachers improve their adaptability by increasing learning engagement. By maintaining sustained learning engagement, pre-service teachers can create an optimal learning environment, integrate more effectively into educational settings, and respond flexibly to various challenges they encounter, thus better adapting to university life. This enhancement of adaptability instills confidence in them as they prepare for future professional environments while simultaneously fostering the evolution of their professional identity. A strong capacity for adaptability may enhance self-efficacy, subsequently impacting professional identity. Pre-service teachers with good adaptability are better equipped to navigate the various challenges encountered during the learning process. At the same time, pre-service teachers with strong adaptability can effectively buffer the teaching pressure after entry ([19]). They effectively leverage support resources, such as assistance from mentors and peers, thereby accumulating successful experiences that further bolster their self-efficacy ([55]). This dynamic is conducive to establishing a positive teacher professional identity. Conversely, if pre-service teachers struggle to manage the pressures and challenges associated with their studies and personal lives, they may experience feelings of frustration. Such negative experiences can adversely affect their professional identity, consistent with findings from existing relevant research ([73]).

### 5.3. The Mediating Role of Self-Concept

These results indicate a positive association between learning engagement and professional identity, with self-concept playing a mediating role. Additionally, students demonstrating higher levels of learning engagement tend to exhibit better academic outcomes. A positive learning experience encourages these individuals to develop a constructive self-concept, which subsequently enhances their sense of professional identity within the teaching profession. Bandura’s self-efficacy theory posits the pivotal role of an individual’s belief in their capabilities in shaping behavioral transformation ([5]). By increasing their learning engagement, pre-service teachers accumulate professional knowledge and teaching experience, thereby bolstering their self-efficacy and facilitating the positive development of self-concept. Furthermore, self-efficacy, as a crucial component in self-concept, directly influences one’s behavioral choices, levels of effort, and persistence ([85]). For pre-service teachers, their self-concept may encompass self-assessments regarding their teaching abilities, communication skills, sense of responsibility, and other related attributes. These self-evaluations can significantly impact their perception of suitability for the teaching profession as well as their commitment to pursuing it. Students who maintain a positive self-concept and believe they possess the requisite professional knowledge, skills, and qualities expected of teachers are likely to bolster their confidence in teaching roles; this enhancement contributes positively to their professional identity. Conversely, if an individual’s self-concept is predominantly negative, they may face greater challenges in their academic pursuits and experience doubts about their appropriateness for a career in education; such circumstances could cause a decline in professional identity. Previous studies have corroborated these findings ([32]; [46]).

### 5.4. The Sequential Mediating Role of Adaptability and Self-Concept

The current investigation found that learning engagement can be linked to professional identity through the enhancement of adaptability and self-concept. Findings reveal that pre-service teachers’ learning engagement impacts professional identity through the sequential mediation involving adaptability and self-concept. When pre-service teachers demonstrate higher learning engagement, they may have stronger adaptability, which promotes self-concept and ultimately enhances their professional identity. This phenomenon can be elucidated by the career decision-making model introduced by [52] ([52]), which emphasizes that through continuous experiential learning and activities, individuals acquire more information and experience while cultivating foresight regarding field trends, thereby enabling them to make relatively rational career decisions. The current study further enhances our understanding by indicating that adaptability and self-concept are both critical predictors of professional identity. This novel perspective suggests that if pre-service teachers lack adequate adaptability and a well-rounded self-concept, their learning engagement may not significantly enhance their professional identity.

### 5.5. Research Implications

This study, utilizing a quantitative research approach, examines the relationship between pre-service teachers’ learning engagement and professional identity, as well as the successive intermediary influences of adaptability and self-concept. Theoretically, the empirical analysis reveals the complicated relationships among learning engagement, adaptability, self-concept, and professional identity, thereby laying a theoretical groundwork for subsequent research. Particularly, delving into sequential mediation enhances our comprehension of the mechanisms that underpin professional identity. Moreover, the methodology utilized in this investigation has set a benchmark for analogous studies, which helps promote the in-depth development of quantitative research within the realm of pre-service teacher education. Practically, it presents a fresh perspective on the cultivation of pre-service teachers’ professional identity. It should be emphasized that all the following practical suggestions and related discussions are closely centered on the pre-service teacher group. Firstly, relevant departments should provide multifaceted support to pre-service teachers during their academic life. Universities should create a favorable campus atmosphere for pre-service teachers and enhance their professional emotions. Meanwhile, universities and teacher educators ought to place significant emphasis on understanding and addressing the emotional demands of pre-service teachers by stimulating their enthusiasm for learning and nurturing a passion for careers in education. Secondly, effective intervention approaches should be formulated and carried out. Efforts should be made to improve the adaptability and stress-coping skills of pre-service teachers. Universities can implement mental health education initiatives, offer mental health assistance and competency training, and help pre-service teachers adapt to academic life. If pre-service teachers can effectively manage stress and respond flexibly to various challenges, they are likely to exhibit greater determination in their career choices ([49]). Meanwhile, pre-service teachers are advised to adjust their learning strategies promptly and effectively adapt to academic life through reflective exercises and mindset adjustments. Finally, teacher educators should focus on stimulating pre-service teachers’ self-concept during the teaching process; This may be achieved, for instance, by providing timely and specific positive feedback while assisting pre-service teachers in recognizing their progress, and enhancing teaching self-efficacy through microteaching experiences, internships, and other methods. These opportunities enable pre-service teachers to accumulate successful teaching experiences and reinforce their belief in their ability to fulfill the role of an educator.

## 6. Limitations and Future Research

Based on the support of social cognitive theory and relevant empirical research, this study constructed a sequential mediation hypothesis model, aiming to explore the possible mediating mechanisms of adaptability and self-concept between learning engagement and professional identity. It should be noted that this study is an exploratory one, mainly verifying the theoretical hypothesis path of “learning engagement→adaptability→self-concept→professional identity”. However, we also recognize that this model has several limitations. Firstly, according to the dynamic interaction principle of social cognitive theory, there may be bidirectional or even multi-directional influence relationships among these variables. For instance, pre-service teachers’ professional identity might enhance their learning engagement. This study mainly verified one directional correlation path among the variables, and there are other relationship paths among them. We believe that future research can explore more correlations or causal relationships on this basis.

Secondly, the cross-sectional nature of this study limited the examination of the temporal dynamics of professional identity development. Although we examined pre-service teachers at different phases, ranging from freshman to senior, we did not continuously monitor the variations in these pre-service teachers’ professional identities across different learning stages. In reality, pre-service teachers’ professional identity is a dynamic development process, and only through long-term tracking can the changes in professional identity be understood more comprehensively. Existing studies have indicated that pre-service teachers’ professional identity changes over time, from an initial idealization to a more realistic state ([65]). Additionally, pre-service teachers’ professional identity undergoes changes during the transition from the preparatory phase to the initial teaching stage ([42]). Furthermore, due to the cross-sectional nature of the study, it was impossible to determine causal relationships or the direction of associations. At the same time, this study investigates the pre-service teachers’ learning engagement and professional identity in the context of Chinese culture, which may reduce the universality of the conclusions to individualistic cultures. Therefore, future research should focus on longitudinal designs or cross-cultural comparative studies to verify the applicability of the conclusions in other educational settings.

In addition, when exploring pre-service teachers’ professional identity, emotional factors received inadequate attention. One potential avenue for extending the current research in future studies lies in examining pre-service teachers’ emotional aspects. The emotional factors of pre-service teachers play a key role in the construction of their professional identity ([88]). [89] ([89]) confirmed that the emotions of pre-service teachers affect their learning attitudes and understanding of the teaching profession. Likewise, [56] ([56]) discovered that emotional factors can promote the engagement of teacher trainees in their teaching work. Therefore, by studying the psychological impact of emotional factors on the professional identity of pre-service teachers, practical suggestions can be provided for teacher education and training.

Finally, although we controlled for demographic variables, such as gender, grades, several key confounding variables, such as GPA, socioeconomic status, and internship experience of some senior pre-service teachers, were not measured. The failure to control for these variables may have weakened the reliability of the research results, and they should be further considered in future studies.

Addressing these limitations, future research should focus on additional critical elements shaping pre-service teachers’ professional identity and should address more nuanced issues such as their career intentions and aspirations. Given the limitations of our study concerning data results and literature experience, we employed a sequential mediation effect exploration model to clarify the relationships among learning engagement, adaptability, self-concept, and professional identity. Therefore, we recommend that future research investigate more sequential relationships to enhance or complement our findings. Furthermore, we found that specific aspects of learning engagement, such as support from teachers, support from peers, and teacher education courses, can significantly enhance pre-service teachers’ professional identity. Future studies could delve into more specific aspects of learning engagement to propose targeted and feasible practical recommendations. As suggested by [43] ([43]) and [90] ([90]), emotional engagement and interactions with peers might contribute to establishing pre-service teachers’ professional identity. Both universities and teacher educators serve vital roles in facilitating dialogue around knowledge formation with oneself, others, and the broader world. Future research should explore pre-service teachers’ professional identity from a wider array of dimensions and structures.

## 7. Conclusions

The present study investigates pre-service teachers in Chinese universities, exploring the connection between their learning engagement and professional identity, as well as the interplay and influence mechanisms of adaptability and self-concept within this context.

The research indicates that (1) pre-service teachers’ learning engagement was significantly correlated with professional identity; (2) adaptability serves as a mediator linking pre-service teachers’ learning engagement to their professional identity; (3) self-concept serves as a mediator linking pre-service teachers’ learning engagement to their professional identity; (4) adaptability and self-concept play sequential mediating roles linking pre-service teachers’ learning engagement to their professional identity. This study is based on a sample of pre-service teachers from Chinese universities. The research findings indicate that in the context of Chinese culture, teacher educators ought to attach great importance to pre-service teachers’ learning engagement. Through the establishment of a positive learning environment, guiding them to deal with challenges, and continuously improving their adaptability, a more stable self-concept can be formed, thereby enhancing their professional identity.

## Figures and Tables

**Figure 1 behavsci-15-00881-f001:**
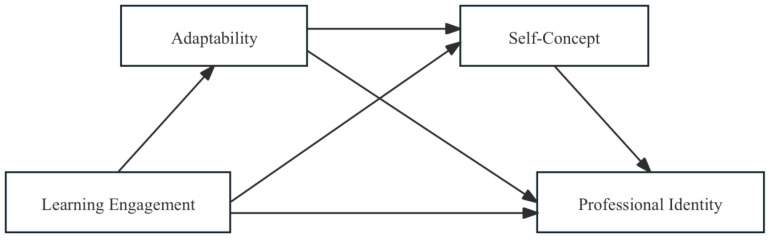
Hypothesized research model.

**Figure 2 behavsci-15-00881-f002:**
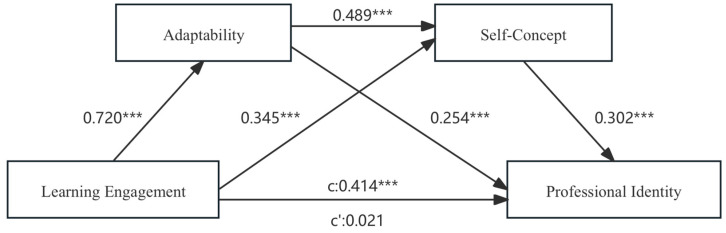
The sequential mediating model (Note: *** indicates *p* < 0.001).

**Table 1 behavsci-15-00881-t001:** Sample demographics (*n* = 632).

Variable	Classification	Frequence	Percentage (%)
Gender	Male	114	18.0
	Female	518	82.0
Residence	Rural	329	52.1
	Urban	303	47.9
Cultivation type	Public-funded	395	62.5
	Non-public-funded	237	37.5
Grade	Freshmen	273	43.2
	Sophomores	117	18.5
	Juniors	182	28.8
	Seniors	60	9.5

**Table 2 behavsci-15-00881-t002:** Mean, standard deviation, and the correlation matrix of each variable (*n* = 632).

Variables	M	SD	1	2	3	4	5	6	7	8	9	10	11	12	13	14
1. Course study	3.691	0.744	1													
2. Deep learning approach	3.697	0.599	0.477 **	1												
3. Student–faculty interaction	2.977	0.924	0.427 **	0.418 **	1											
4. Peer interaction	3.296	0.809	0.434 **	0.404 **	0.281 **	1										
5. Extracurricular activity	2.351	0.853	0.327 **	0.315 **	0.415 **	0.412 **	1									
6. University belonging	3.888	0.773	0.374 **	0.486 **	0.338 **	0.431 **	0.286 **	1								
7. Academic self-concept	3.785	0.587	0.561 **	0.586 **	0.431 **	0.428 **	0.404 **	0.543 **	1							
8. Professional self-concept	3.674	0.808	0.371 **	0.451 **	0.242 **	0.328 **	0.262 **	0.305 **	0.495 **	1						
9. Academic adaptability	3.792	0.613	0.512 **	0.592 **	0.377 **	0.435 **	0.392 **	0.579 **	0.777 **	0.570 **	1					
10. Psychological adaptability	3.447	0.697	0.400 **	0.472 **	0.369 **	0.393 **	0.316 **	0.622 **	0.566 **	0.423 **	0.686 **	1				
11. Professional aspiration	4.359	0.632	0.270 **	0.304 **	0.184 **	0.219 **	0.121 **	0.376 **	0.364 **	0.347 **	0.391 **	0.336 **	1			
12. Professional will	3.625	0.893	0.240 **	0.135 **	0.131 **	0.187 **	0.072	0.221 **	0.195 **	0.376 **	0.254 **	0.301 **	0.369 **	1		
13. Professional value	4.011	0.668	0.279 **	0.307 **	0.144 **	0.216 **	0.136 **	0.360 **	0.344 **	0.321 **	0.356 **	0.374 **	0.442 **	0.432 **	1	
14. Professional efficacy	4.189	0.616	0.332 **	0.308 **	0.186 **	0.267 **	0.168 **	0.388 **	0.426 **	0.456 **	0.420 **	0.416 **	0.572 **	0.512 **	0.535 **	1

Note: ** indicates *p* < 0.01. The same description applies to the table below. M: mean. SD: standard deviation.

**Table 3 behavsci-15-00881-t003:** Mediating analysis results (*n* = 632).

Outcome Variable	Predictor Variables	R^2^	F	β	t
Adaptability	Constant	0.509	92.329 ***	0.883 ***	7.437
	Learning engagement			0.815 ***	25.259
Self-concept	Constant	0.607	120.172 ***	0.738 ***	6.846
	Learning engagement			0.379 ***	9.501
	Adaptability			0.474 ***	13.633
Professional identity	Constant	0.257	30.894 ***	2.800 ***	21.116
	Learning engagement			0.426 ***	11.830
Professional identity	Constant	0.373	41.064	2.269 ***	17.182
	Learning engagement			0.021	0.425
	Adaptability			0.231 ***	4.928
	Self-concept			0.283 ***	5.968

Note: *** indicates *p* < 0.001.

**Table 4 behavsci-15-00881-t004:** Path analysis results (*n* = 632).

	Effect	Boot SE	Boot LLCI	Boot ULCI
Total effect	0.430	0.036	0.355	0.497
Direct effect	0.021	0.050	−0.078	0.121
Total mediation effect	0.405	0.046	0.316	0.498
Ind 1	0.188	0.043	0.106	0.272
Ind 2	0.107	0.025	0.061	0.160
Ind 3	0.109	0.024	0.065	0.159

Note: Boot SE: bootstrap standard error. Boot LLCI: bootstrap lower-limit confidence interval. Boot ULCI: bootstrap upper-limit confidence interval.

## Data Availability

The data that support the findings of this study are available from the corresponding author upon reasonable request.

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
