# Peer review of "Learning Engagement and Professional Identity Among Pre-Service Teachers: The Sequential Mediating Role of Adaptability and Self-Concept"

_behavsci, 2025, doi:10.3390/bs15070881_

Round 1

Reviewer 1 Report

Comments and Suggestions for Authors

Thank you very much for writing this paper – I enjoyed reading it and I think you have done some really valuable and interesting research here. It made me think about my own student teachers and this led me to the conclusion that you need to add some information here, to make this really impactful to an international readership, which this paper could have. My suggested amendments below are mainly to make this more accessible to an international audience, who will need to understand your context in order to draw comparisons with their own. I think with some minor tweaks, this paper could have a much bigger impact on the discipline worldwide. I look forward to seeing a revised manuscript! Thanks again. 

Abstract:

Some of the sentence construction here is difficult for the reader – the multiple semi-colons in sentence lines 13 – 18 for example. The abstract would benefit from a re-write to make the study more accessible to general readers in education, as this is going to drive up the downloads of your paper and ultimately make it impactful.

Introduction:

Line 26 – give some examples of this research area

Line 38 /39 – not sure you have made enough of a case to claim that identity is crucial in ensuring educational quality – this perhaps merits a paragraph of it’s own and a deeper dive into the research literature if you want to make the claim.

You are citing some really dated literature – some is undoubtedly still relevant, but there are more recent examples you could be using which would be more credible, given how much the sector has changed in the last twenty years. Have a look at the Journal for the Education of Teachers and Teaching and Teacher Education.

In the introduction, it would be good to spell out why this is important work – globally, there is a teacher recruitment and retention crisis in many countries, and there is much interest in recruiting to teacher education programmes from many governments (Australia, UK, Germany for example).

Literature Review

It would be very useful if you referenced the countries studies have been undertaken in, to give the global context.

All of your participants were learning in China – but you haven’t told us anything about the Chinese teacher education system. For this work to be relevant to other countries, it would be really useful if the reader had this information so they could compare with their own systems.

Research methods

Were the participants the students of the researchers? How do you know that they genuinely had the choice not to participate, if they were randomly selected and instructed to fill in the survey? Were there some power issues here?

Author Response

Please see the attachment [Response to Reviewer 1 Comments].

Reviewer 2 Report

Comments and Suggestions for Authors

Thank you for the opportunity to read this comprehensive study, which offers insights drawn from a different cultural context. This cross-cultural perspective is valuable in broadening our global understanding within the research community.

The manuscript is grounded in current literature on professional identity, and it references several prominent scholars in the field. However, the review could be further strengthened by incorporating more comprehensive and recent syntheses of the literature, such as OECD reports, as well as theoretical frameworks that refine the discourse on professional identity. Doing so would enhance the critical framing and conceptual clarity of the study, particularly with respect to the lens through which professional identity is examined.

Given that professional identity is the central concept of the study, its theoretical and conceptual treatment should be more robust. While its significance is acknowledged, and relevant research is cited, there is insufficient engagement with the specific dimensions or types of professional identity being studied. A clearer articulation of the characteristics of professional identity would improve both the theoretical contribution and the interpretative strength of the findings.

The study is written in a clear and accessible academic style, and the structure is coherent and consistent with academic standards. The hypotheses are generally derived from the literature review, although this process could be more explicitly articulated.

One critical point that requires attention is the interpretation of the results. While the methodology and stated hypotheses focus on examining relationships between variables, the findings and discussion sections often use causal without sufficient methodological justification. For instance:

  • In line 432, the authors refer to positive correlations.
  • However, by line 434, the findings are stated as causal (e.g., “adaptation has a positive effect on self-image”).
  • Similarly, in lines 475, 518, and 554, several factors are described as having direct effects on professional identity or on one another.

These interpretations suggest a causal relationship that is not supported by the cross-sectional design of the study. If the study design does not allow for causal inference, the language in the results and conclusions should reflect this limitation and instead describe associations or correlations.

Relatedly, I recommend refining the language in the conclusion to better align with the methodological constraints of the study. Greater caution is warranted in claiming effects, especially given that the study is not longitudinal and examines a specific point in time among a defined population. The claim (lines 590–603) that identity should not be studied at a single time point is noted, but given that this is the design employed, the limitations should directly address the implications of using a cross-sectional approach rather than suggesting alternative designs that were not pursued.

Recommendations for Revision:

  1. Clarify the conceptual framework of professional identity, specifying its dimensions and relevance to the study's aims.
  2. Differentiate clearly between correlation and causation in both the findings and discussion sections.
  3. Reframe the conclusions using language that reflects the study's methodological boundaries.
  4. Strengthen the limitations section with specific and relevant reflections on the design and scope of the study, rather than general or speculative comments.

Overall, the study addresses an important and timely topic, and with these revisions, it has the potential to make a meaningful contribution to the literature on professional identity.

Author Response

Please see the attachment [Response to Reviewer 2 Comments].

Reviewer 3 Report

Comments and Suggestions for Authors

I have reviewed the manuscript "Learning Engagement and Professional Identity among Pre-service Teachers: The Sequential Mediating Role of Adaptability and Self-concept." Although the topic is relevant to this journal, several critical issues require the authors' consideration and substantial revision before further consideration:

1. The authors stated: "Building on these considerations, this study aims to systematically address the limitations identified in previous studies" (Lines 77–78). However, the authors fail to explicitly define or clarify which specific limitations from prior literature are targeted. This critical omission progressively diminishes the theoretical grounding and scholarly relevance of the research, as a clearly articulated research gap is essential for demonstrating originality and significance.

  1. This fundamental flaw significantly limits the theoretical rigor. The authors propose a sequential mediation model (learning engagement → adaptability → self-concept → professional identity) without adequately explaining why adaptability should logically precede self-concept. This lack of clear theoretical foundation results in ambiguity regarding the directionality and interrelationships among the constructs, diminishing interpretability and weakening the credibility of the authors’ claims. See also my comment in point 5.
  2. The manuscript lacks explicit justification for the chosen sample size (n=632) and sampling procedure. Moreover, combining pre-service teachers across diverse developmental stages (freshman to senior) without addressing developmental differences weakens internal validity (Lines 293–294).
  3. There is insufficient detail on how measurement instruments, such as the Learning Engagement Scale, were refined and validated. Detailed reporting on exploratory and confirmatory factor analyses, reliability checks, and measurement invariance across groups is essential for ensuring measurement rigor (Lines 311–313).
  4. The definitions of adaptability and self-concept overlap significantly (Lines 134–135, 181–182). High correlations (e.g., r = .777 between academic self-concept and academic adaptability) further indicate potential redundancy, challenging the discriminant validity and statistical integrity of the findings. I have similar argument in point 2 about a lacking of sold theoretical framework in the current manuscript. One may doubt the implications of this kind of exploratory study to theory and practice. In other words, why do we need such a study? The authors needs to reconsider their literature review and present a solid theoretical framework/an integrated theoretical framework for supporting convincingly their hypothesis as well as their interpretations later on.
  5. Practical implications are broadly generalized beyond pre-service teachers without adequate justification or empirical support (Lines 86–89). Such generalizations weaken the manuscript’s practical significance. Basically, these participants remain in higher education (teacher education). The authors oversimplified the complicated situations in frontline teaching/overclaimed results implications of their study.
  6. Key potential confounders, such as GPA, and socioeconomic status, and prior/internship teaching experience for some senior grade pre-service teachers were not considered. This omission critically undermines the internal validity and reliability of causal inferences (Lines 418–420). The authors should explain and recognise these limitations
  7. The cross-sectional design limits the ability to establish developmental relationships implied by the mediation model (Lines 592–594). Additionally, generalizations based solely on data from Chinese universities, without adequate cross-cultural reflection, limit external validity and applicability of findings (Lines 641–643).
  8. Other formatting issues for the authors to consider in order to align fully with APA (7th edition):

e.g.,

  • Consistently italicize journal titles and volume numbers, keeping issue numbers in parentheses without italics.
  • Use sentence-case capitalization for article titles and proper-case for journal titles.
  • Consistently format DOIs using the URL format (e.g., https://doi.org/xxx) without the "doi:" prefix, except this is the requirement of this journal.
  • Check for and remove repetitive errors.
  • Ensure consistent spacing and punctuation across all references.

Addressing these concerns (particularly points 1 to 8) through substantial revision is essential for improving the manuscript's theoretical rigor, methodological validity, and overall scholarly impact.

Comments on the Quality of English Language

Generally good, this manuscript can be further edited based on the journal's specific requirement to English quality

Author Response

Please see the attachment [Response to Reviewer 3 Comments].

Round 2

Reviewer 2 Report

Comments and Suggestions for Authors

The authors have thoroughly and directly addressed all the comments. The accompanying response letter is very detailed, providing focused replies and incorporating the revised wording in accordance with the reviewers’ remarks. I find that the manuscript is now well-prepared and suitable for publication

Author Response

Comments:

The authors have thoroughly and directly addressed all the comments. The accompanying response letter is very detailed, providing focused replies and incorporating the revised wording in accordance with the reviewers' remarks. I find that the manuscript is now well-prepared and suitable for publication.

Response:

Thank you very much for your recognition and affirmation of our revision work! Your professional and meticulous review comments have provided crucial guidance for the improvement of the manuscript, significantly enhancing the rigor of the research and the accuracy of the expression.

We attach great importance to every suggestion you have made and have implemented them one by one during the revision process to ensure a clearer logical structure and more solid evidence. The overall quality of the manuscript has been significantly improved, which is inseparable from your meticulous guidance.

Thank you again for your time and effort in reviewing our manuscript. We are deeply honored to receive your positive feedback that it is "suitable for publication". We will continue to maintain academic rigor and improve the quality of our research in the future. We look forward to the manuscript moving smoothly to the next stage.

Reviewer 3 Report

Comments and Suggestions for Authors

While I appreciated the authors’ revisions, at least some were trying to address my previous comments. I still found the path model, though statistically informative, appears largely exploratory in nature. It lacks grounding in a well-established theoretical framework that would justify the hypothesized sequence of mediation (i.e., Adaptability → Self-Concept → Professional Identity). The treatment of adaptability and self-concept as independent, linearly ordered constructs oversimplifies their relationship, particularly in light of reciprocal causation principles central to Social Cognitive Theory. Moreover, the absence of contextual moderators or feedback mechanisms further limits the model’s capacity to capture the dynamic, developmental nature of professional identity formation. These factors collectively suggest that the study may still be in an exploratory stage rather than testing a mature, deductively derived model.
Below are some observations on the current version and comments for the authors’ consideration:
Lines 23-25 “Teacher professional identity has consistently been a prominent focus in teacher education research (Li & Khairani, 2025; Banegas, 2022; Gholami, et al., 2021; Rodrigues & Mogarro, 2019; Timostsuk & Ugaste, 2010).” The term teacher professional identity (TPI) is used by the authors without any conceptual definition or elaboration that should be presented earlier in the introduction than the current form (only until 135-139, the authors defined what is TPI.) Shockingly, the authors listed five citations without any explanation of what each contributes makes the claim appear superficially supported. The statement asserts that TPI is a “prominent focus” but fails to explain why it has received attention or what impact it has on teacher education (e.g., motivation, retention, professional development). The references range from 2010 to 2025, but no explanation is provided about temporal trends in TPI research (e.g., growing attention post-2010; shift in focus from in-service to pre-service).

There are many instances in this manuscript where references are misused or superficially inserted without clear justification or integration into the argument. This raises concerns about the authors’ academic rigor and their ability to engage with the literature meaningfully. Such misuse undermines the credibility of the work and suggests a lack of critical understanding of how to use references effectively to support scholarly claims.

In the below section, I use 2.1 to illustrate and suggest more concretely:
2.1. Learning Engagement and Professional Identity
The discussion of teacher professional identity (TPI) beginning in lines 135–139 is introduced with two definitions—Beijard et al. (2004) and OECD (2022)—but the authors fail to integrate them meaningfully. Beijard defines TPI as “teachers’ cognition of their role as educators and their social recognition as teachers,” while OECD presents it as “a collection of beliefs, emotions, and perspectives about themselves and their role.” These perspectives overlap substantially, yet are presented as separate claims without synthesis, leading to redundancy and conceptual disconnection.

Subsequently, the claim that “as a multidimensional concept of identity, the formation of teacher professional identity is affected by many factors” (line 140) is not well elaborated. The authors list vague elements such as “preferences, goals, values, and concepts” without linking them clearly to any prior definitions or theoretical dimensions. This weakens the coherence of the multidimensionality claim.

The introduction of Van Veen and Sleegers’ (2009) five dimensions—“professional motivation, core work tasks, self-worth cognition, education theme, and understanding of the essence of education work” (lines 143–145)—is abrupt and disconnected from the earlier conceptualizations by Ruohotie-Lyhty & Moate (2016). The authors do not clarify whether these are complementary or alternative frameworks, nor do they integrate them into a coherent model of TPI.

Lines 146–150 further disrupt the logic by shifting immediately to the “measurement problem” and Hanna et al.’s six-dimensional model (e.g., “self-image, motivation, professionalism, self-efficacy, task cognition and job satisfaction”) without a bridging statement. The lack of a conceptual transition from theoretical models to measurement frameworks weakens the argumentative flow.

Moreover, while the authors state that their study adopts Wang’s (2010) four-dimensional model—“professional aspiration, professional will, professional value, and professional efficacy”—to assess TPI in Chinese pre-service teachers (lines 154–156), they provide no justification for choosing this over the more internationally recognized six-dimension model from Hanna et al. This omission leaves the reader unclear on whether the choice was made based on empirical validation, cultural relevance, or pragmatic constraints.

In lines 158–166, the authors introduce Social Cognitive Career Theory (SCCT) to frame the development of TPI, stating that “personal beliefs (self-efficacy, outcome expectations, goals), environmental factors..., and behavioral factors... interact to shape career development paths.” However, this creates conceptual confusion, as SCCT is a developmental model—not a definition of TPI—and its relationship to previously introduced dimensions is left unexplained.
Although the authors cite international studies (Ali Abadi, 2023; Meyer et al., 2023) to support the claim that learning engagement contributes to identity development (lines 167–176), they fail to connect these findings back to the specific TPI dimensions discussed earlier. The empirical findings are treated in isolation rather than contributing to a unified theoretical argument.
The discussion of the Chinese context (lines 177–183) overgeneralizes teacher education structures by claiming that “the duration and depth of practical components... are relatively limited,” without citing sufficient empirical evidence beyond Zhang et al. (2018). This leads to speculative assertions about the impact of engagement on identity development.

Finally, in lines 184–193, the authors imply a mutual influence between engagement and identity—“pre-service teachers can gradually clarify their career directions... through in-depth learning”—yet the causal direction remains vague. The hypothesis is weakly formulated as “may be significantly related,” lacking specificity in terms of direction and effect. 

Key Issues Identified in Lines 289–300: 
The authors incorrectly portray SCT as supporting a linear “behavior → experience → cognition” pathway. SCT is based on the principle of reciprocal determinism—a dynamic interaction among personal, behavioral, and environmental factors (Bandura, 1986, 1997). This mischaracterization oversimplifies the theoretical foundations and undermines the validity of the proposed model.

The text conflates self-efficacy—defined by Bandura as one’s belief in their capability to perform specific tasks—with self-concept, a broader, multidimensional construct encompassing general self-evaluations (e.g., academic, social, and physical domains; Marsh & Shavelson). These constructs are theoretically distinct and should not be used interchangeably without clear operational definitions or justification.

The assertion that adaptability serves as the “foundation” for self-concept lacks theoretical and empirical support. It neglects the influence of other well-established antecedents of self-concept formation, such as social comparison, external feedback, and prior academic achievement. Furthermore, the claim that self-concept mediates between adaptability and professional identity is speculative and not backed by robust empirical evidence.

Although Bandura’s foundational works are cited (1986, 1997), the authors fail to extract or explain the specific mechanisms relevant to their proposed mediation model. The absence of direct conceptual linkages or quoted theoretical principles renders the use of these references superficial and unconvincing.

Round 3

Reviewer 3 Report

Comments and Suggestions for Authors

I appreciate the authors’ efforts to revise the manuscript and address several of my prior concerns. However, a number of key issues remain only partially resolved.

Specifically, the manuscript still does not sufficiently justify the linear mediation sequence (Adaptability → Self-concept → Professional Identity) with reference to established theoretical frameworks. I recommend that the authors explicitly articulate how Social Cognitive Theory or alternative models (e.g., reciprocal or contextual models of identity development) underpin their hypothesized pathways, and more transparently acknowledge the exploratory nature (this is what the authors replied to my previous concerns) and limitations of their chosen structure. Including a conceptual diagram that maps each construct to relevant theory would help clarify these relationships. A related question to the authors, if SCT is not your conceptual framework, why it is suitable to explain your results. I would suspect you tended to use a kind of data-driven approach, meaning based on your results, you explore which conceptual framework would help you explain the results, instead of providing a clear conceptual framework at the very beginning. This "bad" practice will limit implications of your study findings, both theoretically and practically. 

Additionally, while there is improved description of references and some effort to link them to arguments, the manuscript would benefit from a more critical and integrative synthesis of prior literature. Rather than citing studies in isolation (in other words, synergise what you cited, not just put references there at a surface level), please systematically explain how each referenced work informs your conceptual model, variable selection, and cultural context—ideally using direct quotations from the revised text to demonstrate these improvements. Finally, where decisions diverge from international conventions (such as selecting Wang’s four-dimensional TPI model), offer a concise, evidence-based justification and explicitly discuss implications for generalizability. These steps will strengthen the manuscript’s theoretical rigor, clarity, and value to the field.
